# ADAM10 Plasma and CSF Levels Are Increased in Mild Alzheimer’s Disease

**DOI:** 10.3390/ijms22052416

**Published:** 2021-02-28

**Authors:** Izabela Pereira Vatanabe, Rafaela Peron, Marina Mantellatto Grigoli, Silvia Pelucchi, Giulia De Cesare, Thamires Magalhães, Patricia Regina Manzine, Marcio Luiz Figueredo Balthazar, Monica Di Luca, Elena Marcello, Marcia Regina Cominetti

**Affiliations:** 1Department of Gerontology, Federal University of São Carlos, São Carlos 13565-905, Brazil; izabelavatanabe1@gmail.com (I.P.V.); rafaelaperoncardoso@gmail.com (R.P.); marinagerontologia017@outlook.com (M.M.G.); patricia_manzine@yahoo.com.br (P.R.M.); 2Department of Pharmacological and Biomolecular Sciences, Universitá Degli Studi di Milano, 20133 Milan, Italy; silvia.pelucchi@unimi.it (S.P.); decesaregiulia1@gmail.com (G.D.C.); monica.diluca@unimi.it (M.D.L.); 3Department of Neurology, University of Campinas, Campinas 13083-887, Brazil; thamiresncm@hotmail.com (T.M.); mbalth@unicamp.br (M.L.F.B.)

**Keywords:** ADAM10, biomarker, Alzheimer’s disease

## Abstract

ADAM10 is the main α-secretase that participates in the non-amyloidogenic cleavage of amyloid precursor protein (APP) in neurons, inhibiting the production of β-amyloid peptide (Aβ) in Alzheimer’s disease (AD). Strong recent evidence indicates the importance of the localization of ADAM10 for its activity as a protease. In this study, we investigated ADAM10 activity in plasma and CSF samples of patients with amnestic mild cognitive impairment (aMCI) and mild AD compared with cognitively healthy controls. Our results indicated that plasma levels of soluble ADAM10 were significantly increased in the mild AD group, and that in these samples the protease was inactive, as determined by activity assays. The same results were observed in CSF samples, indicating that the increased plasma ADAM10 levels reflect the levels found in the central nervous system. In SH-SY5Y neuroblastoma cells, ADAM10 achieves its major protease activity in the fraction obtained from plasma membrane lysis, where the mature form of the enzyme is detected, confirming the importance of ADAM10 localization for its activity. Taken together, our results demonstrate the potential of plasma ADAM10 to act as a biomarker for AD, highlighting its advantages as a less invasive, easier, faster, and lower-cost processing procedure, compared to existing biomarkers.

## 1. Introduction

The amyloidogenic cleavage of the amyloid precursor protein (APP) results in the aggregation and accumulation of β-amyloid (Aβ) in senile plaques, which lead to inflammation, tauopathies, and consequent synaptic loss characteristic of Alzheimer’s disease (AD) [1]. On the other hand, the non-amyloidogenic cleavage of APP is carried out by α and γ-secretases and prevents Aβ formation [2]. The main α-secretase participating to the non-amyloidogenic cleavage of APP in neurons is ADAM10, a disintegrin and metalloproteinase [3].

Early diagnosis is crucial to improving the prognosis of and the therapeutic approaches to the disease, and several strategies have been employed with this aim, including imaging, cerebrospinal fluid (CSF) testing and, more recently, blood evaluation of specific AD markers. Blood-based AD biomarkers are advantageous over other diagnostic methods owing to several aspects, including their use in non-invasive and inexpensive screening [4]. In this regard, ADAM10 has been identified as an AD biomarker candidate. This protein has been shown to be reduced in the platelets of AD patients compared to cognitively healthy individuals [5,6,7,8,9]. Moreover, platelet ADAM10 levels have been shown to increase throughout cognitively healthy aging [10]. 

ADAM10 acts as a sheddase cleaving several substrates at the plasma membrane, including APP in neurons [11,12], where it is located in both the pre- and postsynaptic compartments in intact brain sections [11]. It was recently demonstrated that only the active form of ADAM10 is expressed at the surface of different cell types, including leukocytes derived from peripheral blood [13]. Moreover, translocation of the negatively charged phospholipid phosphatidylserine (PS) to the outer membrane leaflet is pivotal in order for ADAM10 to exert its sheddase function [14]. On the other hand, plasma-soluble ADAM10 levels were found to be increased early in amnestic mild cognitive impairment (aMCI) and in AD patients compared to cognitively healthy older adults [15]. 

We therefore hypothesized that the higher plasma ADAM10 levels found in aMCI and AD patients would be correlated with less active protein amounts at the membrane exerting sheddase activity. In order to test this hypothesis, we investigated ADAM10 activity in the plasma and CSF of aMCI and mild AD patients. Our findings help to increase our understanding of the regulation of ADAM10 activity, thereby facilitating the development of more suitable AD biomarkers.

## 2. Results

As ADAM10 is a multidomain protein with multiple cleavage sites, it can be present in different sizes and forms (Figure 1A). As such, we analyzed different ADAM10 isoforms using specific antibodies that recognize the C-terminal (ProSci 2051) and N-terminal (Abcam 39153) regions of the protein (Figure 1B,C), respectively. The western blotting analyses carried out with the antibody that recognizes the C-terminal end of ADAM10 revealed a specific band at 60 kDa, in platelets, as expected for the mature full-length form of the enzyme. On the other hand, western blotting experiments of CSF and plasma samples performed using the antibody directed against the N-terminal portion of ADAM10 revealed a specific band at the apparent molecular weight of 50 kDa (Figure 1B). Western blotting analysis of aliquots of 10 and 20 μg of proteins of two different plasma samples revealed that the 50 kDa band is recognized by both ADAM10 antibodies, i.e. the antibody recognizing the C-terminal ADAM10 domain and the antibody against the N-terminal ADAM10 region (Figure 1C). Further Western Blot analysis confirmed that both ADAM10 antibodies recognize the band at 50 kDa in plasma and in CSF (Appendix A). To assess whether such ADAM10 form was bound to vesicles, an aliquot of CSF was centrifuged at 100,000× *g* for 70 min to remove extracellular vesicles [16]. As shown in Appendix A, the centrifugation doesn’t affect the detection of ADAM10 in CSF, suggesting that the form at 50 kDa is not associated to vesicles (Appendix A).

Table 1 shows the sociodemographic, clinical, and neuropsychological variables of subjects in the control, aMCI, and mild AD groups. The participants were paired according to sex, age, and education level. In CSF, t-tau and p-tau levels significantly increased with disease progression, whereas Aβ levels decreased; however, the differences were not significant among the groups. Although not significant, ADAM10 levels showed a tendency to increase in the aMCI group and a significant increase in the AD group was observed in both the CSF and plasma (Table 1). 

Figure 2 shows representative images of the western blotting assay and quantification from CSF and plasma samples. The aMCI group showed no significant differences in ADAM10 levels when compared to controls in both CSF and plasma samples. However, ADAM10 levels of the mild AD group were significantly increased compared to the control in the CSF (*p* = 0.02) and plasma (*p* = 0.01), respectively (Figure 2A,B).

To assess ADAM10 proteolytic activity we set up an in vitro assay using a fluorogenic substrate. To this, we took advantage of SH-SY5Y cells and we performed a subcellular fractionation in order to test the enzymatic activity of ADAM10 in different compartments. We used a lysis buffer containing digitonin, a steroidal saponin that permeabilizes plasma membrane without disrupting the membrane of cellular organelles, as cholesterol composition of these membranes is lower [17,18]. After a centrifugation, we obtained the fraction S1 that contains plasma membrane-inserted proteins, as the mature form of ADAM10 and APP, and cytosolic proteins as GAPDH (Figure 3A). Then, the pellet was lysed with a buffer containing Nonidet P-40 to release the proteins from organelles except the nucleus, as shown by the presence of the Endoplasmic Reticulum protein Calnexin and the absence of the histone H3 protein in the S2 fraction. In this fraction we detected both the immature and the mature form of ADAM10 and APP (Figure 3A). The in vitro assay carried out with the fluorogenic substrate of ADAM10 revealed that its activity is higher in the S1 than in the S2 fraction, as expected from the distribution of the mature and immature forms of the enzyme (Figure 3B).

ADAM10 activity was evaluated in different biological materials (CSF, plasma, SH-5YSY neuroblastoma cells, and platelets) (Figure 3C). ADAM10 presented no activity on the fluorogenic substrate in CSF and plasma samples of control individuals and in those with aMCI and AD. On the other hand, in platelets, SH-5YSY cells, and in the positive control (recombinant ADAM10), ADAM10 activity increased over time (Figure 3C). The presence of a specific inhibitor for ADAM10 (GI254023X) in the platelets sample completely abolished ADAM10 activity, thus demonstrating that the majority of the activity in samples was due to ADAM10 (Figure 3C).

## 3. Discussion

Blood-based AD biomarkers are highly desired in clinics for more accurate diagnoses, especially if they are able to detect the disease in an early phase. In this study, we provided additional evidence that ADAM10 could be a blood-based AD biomarker candidate, demonstrating that the protein has higher levels in the plasma of patients with mild AD, compared to cognitively healthy older adults. This result is in line with the changes in ADAM10 levels found in CSF, thus connecting circulatory and central nervous system findings. Moreover, we showed that the 50 kDa ADAM10 form found in plasma and CSF was unable to cleave a fluorogenic ADAM10 substrate and is, therefore, inactive. On the other hand, ADAM10 was active in platelets and neuroblastoma cells. 

We and other groups have reported that platelet levels of ADAM10 are reduced in AD [7,8,19,20] and that this reduction is correlated with worse performance in cognition, as measured by the clock drawing test [5] and MMSE scores [6]. Remarkably, the levels of ADAM10 in platelets increase in an age-dependent manner, suggesting that it may be required for cognitively healthy aging [10]. As already demonstrated, ADAM10 is abundant on the cell surface and can be detected by staining with monoclonal antibodies, followed by flow cytometric analyses [21]. Moreover, the presence of ADAM10 on the cell surface is crucial for leukocyte migration and their inflammatory recruitment into the alveolar space [22]. The regulatory mechanisms underlying ADAM10 sheddase activation are not fully understood, but it has been demonstrated that the transient exposure of phosphatidylserine is required for the protein to exert its activity [14]. In addition, a recent paper by Seifert et al. [13] showed that ADAM10 must be membrane-bound in order to exert its activity in different cell types and in leukocytes derived from peripheral blood. In addition, the use of the classical inhibitor GI254023X resulted in a considerable reduction of surface-detectable ADAM10 compared to control cells, which were not treated with the inhibitor [13]. 

Here we analyzed ADAM10 activity in different subcellular fractions to set up a method for the assessment of ADAM10 activity. We obtained plasma membrane and cytosolic proteins (S1) and a fraction of membrane-bound proteins from intracellular organelles (S2) from SH-SY5Y cells and tested their activity against a specific ADAM10 fluorogenic substrate. Total homogenate and S1 fractions were the most active in cleaving the ADAM10 substrate. The results are in accordance with our hypothesis that ADAM10 is active at the plasma membrane. Considering that digitonin is the main component of the buffer used to isolate S1 fraction and this saponin permeabilizes the plasma membrane by forming pores and leading to its subsequent disruption [17,18], it is reasonable to find active ADAM10 in these fractions. In line with these results, western blotting analyses showed the presence of the mature form of ADAM10 in the S1 fraction. The S2 fraction, containing both the immature and mature forms of ADAM10, was less active in cleaving the ADAM10 fluorogenic substrate. This can be explained since, in addition to cell surface, ADAM10 cleavage and maturation occur in the trans-Golgi network and in vesicles of the secretory pathway [23,24,25].

In agreement with our study, ADAM10 was shown to be constitutively active in cell-based assays that measure ADAM10-dependent substrate release [26]. Additionally, N-methyl-D-aspartate (NMDA) receptor activation under physiological conditions induces ADAM10 trafficking through association with SAP97 towards postsynaptic membranes and enables APP cleavage at the α-secretase site in rat hippocampal neuronal primary cultures [27]. Furthermore, recent findings point to an important role of tetraspanins (TSPs), a family of transmembrane proteins, in ADAM10 trafficking, maturation, and sheddase activity. It has been shown that ADAM10 has six TSP partners that can positively or negatively regulate its function at the membrane [28,29,30,31]. All this evidence points to the need for membrane-bound localization in order for ADAM10 to exert its sheddase function. 

In platelets, ADAM10 was demonstrated to be responsible for the cleavage of the adhesion-signaling receptor glycoprotein VI (GPVI), induced by physiologic agonists and by the shear stress provoked by circulation in the vascular tissue [32]. Importantly, an ADAM10-sensitive fluorescence resonance energy transfer (FRET) sensor was able to detect basal ADAM10 activity on resting platelets, supporting the concept that platelets express mature ADAM10 on their surface [32,33]. However, to the best of our knowledge, both the presence and activity of ADAM10 in plasma and CSF have not been demonstrated yet. The source of inactive soluble ADAM10 present in CSF is not known but could be attributed to neuronal death [34], and that in plasma could be due to exosome trafficking. 

However, except for the study presented here, only active ADAM10 forms have been reported in plasma so far [35,36,37]. ADAM10 can undergo ectodomain shedding driven by other members belonging to the ADAM family, such as ADAM9 [38]. Even though ADAM10 shedding has been considered as a mechanism that is able to down-regulate its activity, degradome analyses of soluble ADAM10 have resulted in distinct cleavage products on APP isoforms, demonstrating that the soluble ADAM10 derived from murine cardiomyocytes is active [39]. Therefore, a mechanism of alternative splicing that produces a different ADAM10 isoform could explain its inactivity in plasma and CSF samples. Alternatively, a regulation mechanism could be in place, performed by tissue inhibitors of metalloproteinases (TIMPs), which are endogenous inhibitors of metalloproteinases (MMP) [40] and ADAMs [41], therefore inactivating the plasmatic ADAM10 isoform. Mass spectrometry analyses of the ADAM10 sequence would help to identify this isoform and should be a matter for further studies. 

The increase in ADAM10 in the plasma of AD patients was also found in a recent study using a highly sensitive microfluidic platform to detect ADAM10. The only difference was that the sensor, due to its ability to detect very low levels of the protein (on the order of femtograms), was able to reveal differences in ADAM10 levels as early as in aMCI patients, compared to cognitively healthy participants [15]. 

Interestingly, Sogorb-Esteve and colleagues [34] reported that ADAM10 levels were reduced in the CSF of patients with AD compared to healthy controls, which is not in agreement with our findings. These apparently conflicting results can be explained because the authors used immunoprecipitation to detect ADAM10 forms, which is highly sensitive and may be the reason for the identification of the other 80- and 55-kDa ADAM10 isoforms in the CSF, whereas we only found the 50-kDa isoform. Moreover, the lower ADAM10 levels in CSF found by Sogorb-Esteve et al. were revealed using gradient ultracentrifugation, which is different from our study, in which we simply used crude CSF, without any previous preparation. In addition, our sample consisted of patients with aMCI and mild AD, in which the Aβ levels were slightly lower than the levels found by the authors. However, the levels of p-tau and especially t-tau in our study were very low compared to the findings of Sogorb-Esteve and colleagues, confirming that our sample was composed of patients in the early stages of the disease, according to the qualified cutoff values reported for a Brazilian cohort [42]. Whether the levels of CSF and plasma 50-kDa ADAM10 change according to disease progression is a question raised from these observations and should be further investigated.

Importantly, a recent 3-year follow-up longitudinal work from our group revealed the predictive effect of increased ADAM10 plasma levels to diagnose impaired cognition, as supported by the decrease in the MMSE score values in the follow-up, which seems to be more significant in those with normal MMSE at baseline. On the other hand, in patients with already altered MMSE scores at baseline, ADAM10 did not act as a predictor of worse cognition in the follow-up assessment [43]. In this regard, the evaluation of ADAM10 plasma levels in patients with suspected cognitive decline may allow early interventions that could retard or even prevent the onset of AD. Hence, our results may support the complementary clinical use of this biomarker, in addition to the classical AD biomarkers. More studies need to be performed using a larger number of participants and investigating the levels of plasma ADAM10 in different dementias in order to confirm the specificity of this marker. Ultimately, however, this study offers important contributions to our understanding of the biology of ADAM10, as well as to the early diagnosis of AD. 

Taken together, our results agree with the hypothesis that the 50-kDa isoform that we found to be elevated in CSF and plasma in AD patients corresponds to an inactive form of ADAM10, as demonstrated by the activity test. Therefore, in this study, we demonstrate the potential of plasma ADAM10 to act as a biomarker for AD and highlight the importance of standardizing the experiments and choosing the appropriate antibodies for its detection. ADAM10 could be a complement to the well-established, although expensive, CSF and imaging biomarkers for AD. Considering that obtaining plasma requires fewer steps compared to platelets, this could be an easier method to of detecting this biomarker.

## 4. Materials and Methods

### 4.1. Patients

The study was approved by the ethics committee of the University of Campinas (n.188.021), and all patients signed an informed consent form prior to any procedure. Patients were recruited from the Neuropsychology and Dementia Clinic of the University of Campinas. The participants were diagnosed with aMCI using the core criteria of the National Institute on Aging/Alzheimer’s Association (NIA/AA) [44]. All aMCI participants had a Clinical Dementia Rating (CDR) [45] of 0.5 (with an obligatory memory score of 0.5), identified using a semi-structured interview. In addition, AD and aMCI participants had pathophysiological evidence of AD (which consisted of low Aβ_42_ and high tau protein in CSF) and memory cognitive impairment confirmed by poor performance on an episodic memory test (Rey auditory-verbal learning test, RAVLT). Moreover, aMCI subjects needed to prove the absence of dementia. Global cognitive status was measured using the Mini Mental Status Examination (MMSE) [46].

### 4.2. Blood and CSF Collection

CSF was collected from 22 elderly individuals (7 individuals had aMCI in single and multiple domains, 7 had mild AD, and 8 were cognitively healthy subjects), whereas plasma was collected from 88 participants (28 had aMCI, 31 had AD and 29 were cognitively healthy). Tubes containing sodium citrate solution (3.8%) and glucose (136 mM) were used to collect blood (8.5 mL) by venipuncture. After collection, the tubes were kept at 4 °C during transportation. The tubes were then transported and centrifuged at 1200 rpm for 10 min to obtain platelet-rich plasma (PRP). Subsequently, a new centrifugation was performed to remove the platelets, and the supernatant representing the platelet-poor plasma (PPP) was diluted (1:10) and stored at −80 °C, together with the isolated platelets, for future use.

CSF was obtained from the participants once, by lumbar puncture of the L3/L4 or L4/L5 intervertebral space, using a 25-gauge needle, and collected via a syringe in 12-mL polypropylene tubes (Sarstedt, Nümbrecht, Germany). CSF samples were centrifuged to remove cells at 700 rpm for 10 min at 4 °C and were subsequently aliquoted into 1 mL microtubes (Eppendorf, Hamburg, Germany) and stored at −80 °C until analysis. CSF Aβ_1–42_, total tau, and p-tau proteins were measured using the Inno-Bia Alzbio3 kit (Innogenetics, Gent, Belgium). To assess the association of ADAM10 to extracellular vesicles, an aliquot of CSF was centrifuged at 100,000× *g* for 70 min at 4 °C, to remove extracellular vesicles and cell fragments [16]. The supernatant was collected and loaded onto a SDS-PAGE to carry out Western Blot analysis.

### 4.3. Cell Culture 

Human neuroblastoma cells (SH-SY5Y) were acquired from the Rio de Janeiro Cell Bank (Rio de Janeiro, RJ, Brazil) and maintained in minimum Eagle’s medium (DMEM high glucose; Sigma-Aldrich, St. Louis, MO, USA) containing 10% fetal bovine serum (FBS) and 1% penicillin-streptomycin (Sigma-Aldrich, San Louis, MO, USA). The cells were kept in a humid incubator with 5% CO_2_ and 95% O_2_ at 37°C and maintained in cell culture-specific bottles until they reached the confluency required for use in the proposed assays (cell fractionations).

### 4.4. Subcellular Fractionations

The cell fractionation protocol was based on the study performed by Baghirova and colleagues [17] with modifications. The cells were cultured in two 100-mm plates until they reached confluence, collected, washed with phosphate-buffered saline (PBS), and centrifuged at 5000× *g* for 10 min. The supernatant was discarded, and the cell pellet was resuspended with 400 µL of lysis buffer A, containing 150 mM NaCl, 50 mM HEPES pH 7.4, 25 µg/mL digitonin (D141, Sigma-Aldrich, San Louis, MO, USA). An aliquot of 80 µL of this volume was collected and named “Homo”. The remaining sample was centrifuged for 10 min at 2000× *g*. The supernatant was reserved and called “S1” containing cytosolic and plasma membrane proteins and the pellet was resuspended in 400 µL of lysis buffer B containing 150 mM NaCl, 50 mM HEPES pH 7.4, 50 mM NP-40. The sample was incubated on ice for 30 min and then subjected to a new step of centrifugation for 10 min at 7000× *g*. The supernatant (“S2”) contained mostly proteins from intracellular organelles.

### 4.5. SDS-PAGE and Western Blotting

ADAM10 was evaluated in CSF, plasma, and SH-SY5Y cells. Approximately 20 µg of protein was added to each well of 10% SDS-PAGE gels. After gel electrophoresis, the proteins were transferred to nitrocellulose membranes (Sigma-Aldrich, San Louis, MO, USA) using the mini trans-blot cell transfer system (BioRad, Hercules, CA, USA) for 1 h. The membranes were incubated with rabbit anti-ADAM10 primary antibody (cat. n. 2051, ProSci Poway, CA, USA and cat. n. 39153 Abcam, Cambridge, United Kingdom), followed by incubation with appropriate secondary antibodies (horseradish peroxidase-conjugated goat anti-rabbit, cat. n. 97051, Santa Cruz Biotech Dallas, TX, USA). Ponceau staining revealed the presence of a band of approximately 66 kDa, which corresponded to the protein serum albumin, abundant in the plasma and present in the CSF, which was used as an endogenous control (Figure 2B). In addition to the serum albumin endogenous control, a sample called “young control”, consisting of a suspension prepared from a pool of plasma from young healthy volunteers, was added to each gel to control analytical differences between blotted membranes prepared on different days (inter-assay variation). The measurement of this young control sample was normalized according to the formula (band intensity of ADAM10/band intensity of endogenous control)/band intensity of young control sample. Immunoreactive bands were quantified using Quantity One software (BioRad, Hercules, CA, USA).

The samples obtained in cell fractionation were probed with anti-ADAM10, anti-GAPDH (SC-25778, Santa Cruz Biotech, Dallas, TX, USA) and Anti-Histone H3 (Proteintech, Chicago, IL, USA), anti-Calnexin (cat. n. Adi-SPA-860-F, Enzo Life Sciences, Farmingdale, NY, USA), anti-APP (cat. n. MAB348, Millipore, Burlington, MA, USA) to confirm the isolation of the corresponding fraction, as described below. 

### 4.6. ADAM10 Activity Assay

Approximately 20 µg of protein from CSF, plasma and 150 µg of cell fractions (Homo, S1, S2) were mixed with 10 µM of a fluorogenic peptide (Fluorogenic Peptide Substrate III, cat. n. ES003, R&D Systems; Minneapolis, MN, USA) in reaction buffer composed of 20 mM Tris-HCl and 2.5 µM ZnCl_2_ pH 8.0 to yield a total volume of 100 µL per well or 150 µl for cell fractions assays in a black 96-well plate (Greiner Bio-One, Frickenhausen, Germany). The activity of ADAM10 from platelets and SH-SY5Y neuroblastoma cell lysates was used as a control. The increase in fluorescence intensity was determined with a microplate reader (Enspire, Perkin Elmer, Waltham, MA, USA) at excitation and emission wavelengths of 320 nm and 405 nm (respectively) every 2 min for 60 min, keeping the plate at 37 °C during the measurements. For comparative analysis, values obtained 30 min after initiation of the reaction, subtracted by a blank (buffer with fluorogenic peptide and without protein), were used. Analyses with recombinant ADAM10 (cat. n. 936-AD-020, R&D Systems, Minneapolis, MN, USA) were used to prepare a standard curve with known concentrations, using the same protocol. To check the specificity of the obtained signal, platelets samples were analyzed in the presence of 10 μM of the inhibitor GI254023X (cat. n. 3995, R&D Systems, Minneapolis, MN, USA) in a reaction pre-incubated for 5 min at 37 °C in the dark before adding the fluorogenic peptide.

### 4.7. Statistical Analysis 

To detect sample normality, this study used the Kolmogorov-Smirnov test. Statistical evaluations were performed with the Mann-Whitney U and Kruskal-Wallis tests with a 5% significance level. The figures were built using GraphPad Prism 5 (GraphPad Software, San Diego, CA, USA).

## Figures and Tables

**Figure 1 ijms-22-02416-f001:**
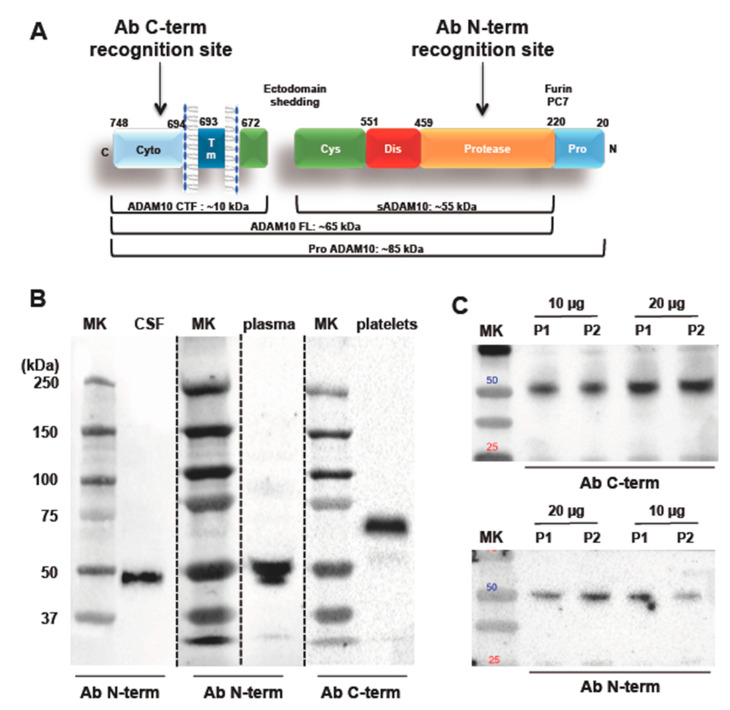
The ADAM10 multimodular structure and its different forms. (**A**) Furin/PC7 proteases cleave Pro-ADAM10, the inactive protein containing a pro-domain (85 kDa), to activate the enzyme during the transit through the Golgi compartment. The full-length active form (ADAM10 FL, 60–65 kDa) is then directed to the cell membrane where it works as sheddase. ADAM10 itself can be subject to ectodomain shedding, a process that leads to the formation of the soluble (sADAM10, 50–55 kDa) and membrane-anchored C-terminal domain (ADAM10 CTF, 10 kDa). Antibody binding sites used in this study to recognize ADAM10 protein. (**B**) Western blotting membranes using the ProSci 2051 antibodies (detection of the C-terminal region) in platelets and Abcam 39153 (detection of the N-terminal region) in plasma and CSF. MK: standard marker. (**C**) Western Blot analysis of 10 μg and 20 μg aliquots of plasma obtained from two control subjects (P1 and P2). Both ADAM10 antibodies recognize a band with an apparent molecular weight of 50 kDa.

**Figure 2 ijms-22-02416-f002:**
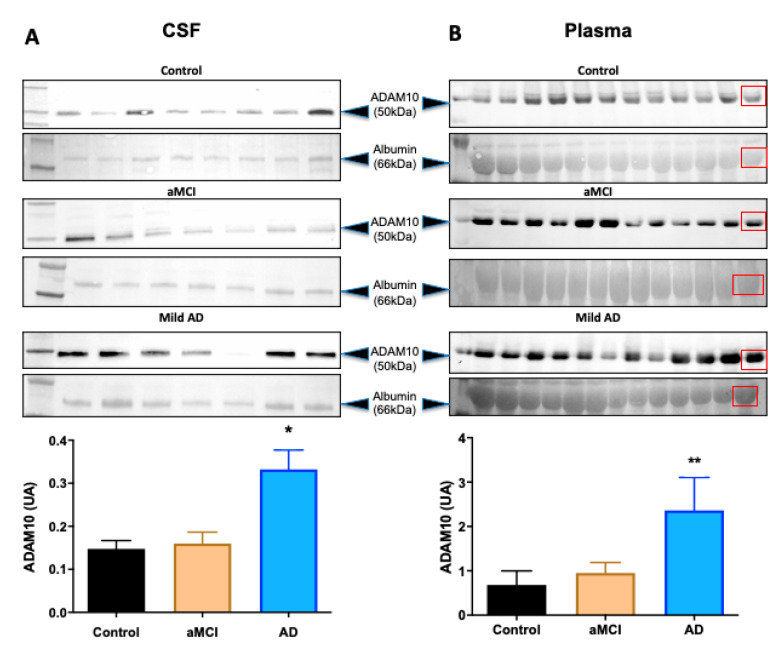
Western blotting assays and ADAM10 protein quantification in CSF and plasma. (**A**) Western blotting and quantification of CSF and (**B**) plasma samples from controls (cognitively healthy), aMCI and mild AD participants probed against anti-N-terminal ADAM10 antibodies, which recognized a 50 kDa form of the protein. Serum albumin (66 kDa) from all participants was used as load control for the quantifications. The last lanes of plasma membranes were loaded with young control samples (red squares). * *p* = *0*.02; ** *p* = 0.01 (control vs mild AD). One-Way ANOVA and Mann-Whitney test. Graph Pad Prism 8.

**Figure 3 ijms-22-02416-f003:**
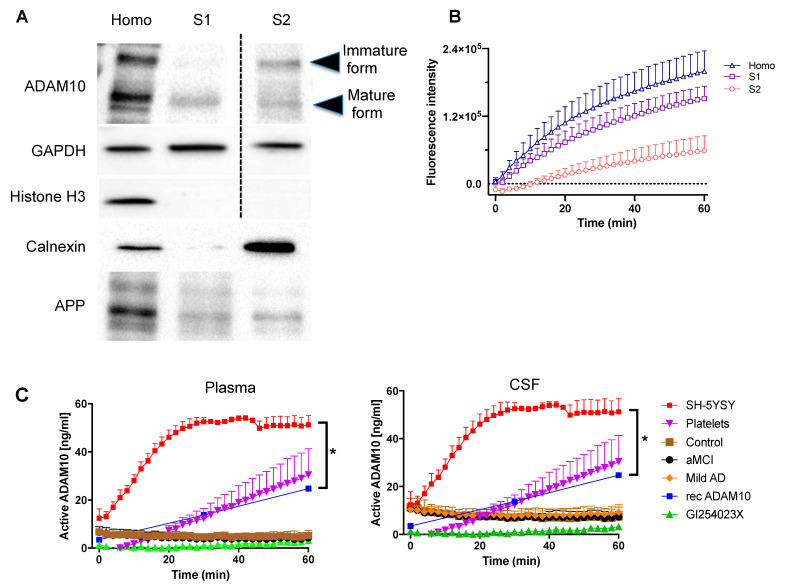
Analysis of ADAM10 activity in SH-5YSY neuroblastoma cells, platelets or CSF samples. (**A**) Western blotting analyses of subcellular fractions of SH-SY5Y human neuroblastoma cells; samples were probed for ADAM10 (immature form 85 kDA, mature form 65 kDa), GAPDH (37kDa) and Histone H3 (17kDa), Calnexin (90 kDa) and APP (106-110 kDa). (**B**) Graph reporting ADAM10 activity monitored for 60 min of reaction in the same fractions described in (**A**). The activity of ADAM10 was evaluated using a fluorogenic peptide as described in methods. (**C**) ADAM10 activity in CSF (rightmost graph) and in plasma (leftmost graph) of control cognitively healthy (control), amnestic mild cognitive impairment (aMCI) and mild Alzheimer’s disease (Mild AD) groups. The ADAM10 activity was also analyzed in SH-5YSY neuroblastoma cells, platelet samples and recombinant ADAM10. A specific inhibitor for ADAM10 GI254023X was incubated with platelets before the addition of substrate to demonstrate that the majority of the activity in samples was due to ADAM10 (* *p* < 0.0001). Two-Way ANOVA and Kruskal-Wallis test. Graph Pad Prism 8.

**Table 1 ijms-22-02416-t001:** Sociodemographic, clinical variables and neuropsychological evaluations of subjects in the groups.

	Control	aMCI	Mild AD
Sex (F/M)	8/0	5/2	7/0
Age	63.7 ± 3.7	71.1 ± 6	72.7 ± 7
Scholarity (years ± SD)	10.2 ± 5.37	7.4 ± 4.4	7.1 ± 4.3
CDR 0 (*n*)	8	7	0
CDR 1(*n*)	0	0	7
CDR2 (*n*)	0	0	0
MEEM (mean ± SD)	28.3 ± 2.3	27 ± 2.1 **	19.5 ± 3.6 ***
CSF t-tau (pg/mL ± SD)	80.3 ± 38.4	103.3 ± 32.8	156.0 ± 86.4 *
CSF p-tau181 (pg/mL ± SD)	35 ± 16.6	38 ± 9.7	62.6 ± 37.5 *
β-amyloid (pg/mL ± SD)	541.7 ± 302.2	530.3 ± 365.2	303.2 ± 107.0
CSF ADAM10 (AU)	0.1475 ± 0.0551	0.1598 ± 0.0707 *	0.3323 ± 0.1185 *^,†^
Plasma ADAM10 (AU)	0.6805 ± 0.3171	0.9512 ± 0.2362 **	2.362 ± 0.7428 **^,†^

* *p* < 0.05 (Control vs mild AD; aMCI vs mild AD); ** *p* < 0.01 (Control vs mild AD; aMCI vs mild AD); *** *p* < 0.001 (Control vs mild AD) in Mann-Whitney U-test; ^†^ One-Way ANOVA (*p* = 0.02 in CSF; *p* = 0.001 in plasma). Graph Pad Prism 5.2.

## Data Availability

Data supporting reported results will be provided upon request.

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
