# Peer review of "ADAM10 Plasma and CSF Levels Are Increased in Mild Alzheimer’s Disease"

_ijms, 2021, doi:10.3390/ijms22052416_

Round 1
Reviewer 1 Report
This study reports that cerebrospinal fluid and plasma of patients with AD show increased expression of an ADAM10 variant but no increase in activity of ADAM10. However, several major concerns need to be addressed.
1) The specificity of the 50 kDa bands for ADAM10 is not proven. One option would be precipitate ADAM10 with one antibody and to detect it via Western blotting with the other antibody. This would at least confirm that the same protein is recognized by both antibodies.
2) The comparative quantification of proteins from different Western blots is problematic. This would require to run a standard on each blot
3) The sample numbers are not specified. It would be interesting to correlate CSF and plasma samples for each patient.
4) Serum albumin levels were used as loading control for control patients. This should also be done for AD patients.
5) S1 was described as supernatant containing cytoplasmic and plasma membrane bound ADAM10, whereas S2 should contain ADAM10 from intracellular compartments. Controls for standard cytoplasmic and membrane bound proteins should be run for each of these fractions similar as it has been done for histone H3 in the lysate.
6) Was ADAM10 from CSF or plasma still bound to cell fragments or vesicles. The vesicles can be removed by ultracentrifugation.
7) The fluorescent substrate assay did not indicate any ADAM10 activity in the CFF or plasma. This could be due to many reasons. Inhibition of ADAM10 activity or degradation of ADAM10 may by one explanation. However, also binding of the peptide and quenching of fluorescence by other proteins could take place. Controls are necessary to further address these issues. For instance, by spiking CSF with recombinant ADAM10 in presence or absence of the GI inhibitor.
Reviewer 2 Report
The authors in manuscript entitled “ADAM10 plasma and CSF levels are increased in mild Alzheimer’s disease” have described the brief overview about the potential of plasma ADAM10 to act as a biomarker for AD, highlighting its advantages as a less invasive, easier, faster, and lower-cost processing procedure.
Authors have tried to convey the clear message about the consented recommendations from the present study, which would be helpful and considered when creating or adapting therapeutic approaches against AD.
The results obtained by the authors are significant and scientifically sound. The paper has been written well and the content is comprehensive and meaningful. This paper will provide a study model for the future path of AD diagnosis biomarkers. The Western blotting assays and ADAM10 activity assay clearly revels the importance and level of ADAM10 protein quantification in CSF and plasma, which depict the role of ADAM10 protein as AD biomarker. Though the paper has a strong framework, the authors surely can add and improve the conclusion part thereby adding more relevant information and dimensions to their work.
To conclude, this paper is a nice compact description about the use of ADAM10 a nice CSF and imaging biomarkers for AD. The quality of the paper is suitable for publication.
Round 2
Reviewer 1 Report
Thank you for the detailed explanations